# Hidden in plain sight: A systematic review of coercion and Long-Acting Reversible Contraceptive methods (LARC)

**Victoria Boydell** [1] *, **Robert Dean Smith** [2], **Global LARC Collaborative (GLC)** [3]

**1** School of Health and Social Care, University of Essex, Colchester Campus, Colchester, United Kingdom, **2** Department of Anthropology and Sociology, Geneva Graduate Institute, Geneva, Switzerland, **3** Health Research, Lancaster University, Lancaster, United Kingdom

* vb21763@essex.ac.uk

## Abstract

In recent years there has been extensive promotion of long-acting reversible contraceptives (LARC) globally to increase access to what is widely considered a highly effective contraceptive method. Yet, despite these efforts, evidence points towards the worrying propensity for LARCS to be associated with coercion. Hence, we undertook a meta-narrative review across nine databases to draw together the heterogeneous and complex evidence on the coercive practices associated with LARC programs. A total of 92 papers were grouped into three metanarratives: (1) law, (2) public health and medicine, and (3) the social sciences. Across disciplines, the evidence supports the conclusion that coercive practices surrounding LARC programs always target marginalized, disadvantaged and excluded population(s). Looking at coercion across disciplines reveals its many forms, and we present a continuum of coercive practices associated with LARC programming. We found that each discipline provides only a partial picture of coercion, and this fragmentation is a knowledge practice that prevents us from collecting accurate information on this subject and may contribute to the perpetuation of these suspect practices. We present this review to address longstanding silences around coercion and LARCs, and to encourage the development of clinical and programmatic guidance to actively safeguard against coercion and uphold reproductive rights and justice.

## Introduction

Long-acting reversible contraceptives (LARCs), e.g., contraceptive methods that require a service provider for insertion and/or removal such as inter-uterine devices, implants and injections, are widely acknowledged to be an integral and empowering part of a comprehensive mix of at least five contraceptive methods that women can choose from and are also considered a cost-effective public health intervention [1]. In 2018, 139.2 million women were using LARCs, approximately 30 percent of all contraceptive users [2]. LARCs represent an increasing proportion of the public sector contraceptive market, constituting 64% in 2016 and increasing

**Data Availability Statement:** Data is available in supplementary information.

**Funding:** This work was supported by the University of Lancaster's Global Challenges Research Funders (UKRI). The funders had no role in study design, data collection and analysis, decision to publish, or preparation of the manuscript. VB received funding for this work.

**Competing interests:** The authors have declared that no competing interests exist.

to 70% in 2020 (excluding injectables) [3]. In 2012, the Population Council, the International Federation of Gynaecology and Obstetrics (FIGO), and the Reproductive Health Supplies Coalition (RHSC) formed a Bellagio Group that argued that over 57 million women would be seeking LARCs by 2020, and advocating for increasing access to these methods was central to achieving universal access to reproductive health [4].This initiative prompted several efforts to expand access to LARCs; such as international donors widening the market share for implants through volume guarantees, and investments in the development and distribution of a revolutionary new LARC, Sayana Press (DMPA-SC) [5].

For some individuals, LARCs may be the ideal contraceptive option to delay, postpone and prevent pregnancies as they have some attractive qualities. LARCs last for a long time (3 months to 5 years), they require less time in terms of the visits to a clinic, and users do not have to think about them daily or when they are having sex. Moreover, they are discrete, so no one can tell if someone is using one. For others, they may be encouraged to use (or not) LARCS on clinical grounds. For a population, moreover, LARCs have been positioned as an efficient means to reduce unplanned pregnancies. For example, for every IUD inserted, a health program can rack up to 4.6 Couple Years of Protection (CYP) from pregnancy compared to 0.06 CYPs that is associated with using the Combined Oral Contraceptive pill [6]. Such investments in preventing 'unplanned' pregnancies can then be linked to many public health, social and economic benefits.

Yet, there is evidence that LARCs have also been associated with coercive practices that compromise people's ability to freely decide about their reproductive futures [7, 8]. Here, we understand coercion as actions and practices that make someone do something they do not want to do through the use of threat, force, intimidation, manipulation, fraud and deceit, as well as through restricting what choices are readily available. As LARCs require a trained professional to be directly involved for its successful application, this involvement inevitably creates an opening for another person to intervene in and/or influence a person's decision. Such practices contravene our most fundamental sexual and reproductive health and rights (SRHR) that encompass the human rights that are relevant for and affect health and wellbeing in respect of sexuality and reproduction. Those rights, which provide the normative basis to SRHR, include but are not confined to "*the right to life, the right to be free from torture, the right to health, the right to privacy, the right to education, and the prohibition of discrimination* [9]." These normative standards mean that women seeking contraceptive care are entitled to make decisions free from coercion and with their active consent.

In the past, there were reported instances of LARCs being forcibly imposed on low-income women, women of color, incarcerated women, and women in the global south. Across the legal, public health and social science literature we have found such accounts in multiple sites including Mexico, Korea. Taiwan, Viet Nam, India and China [10–15]. Moreover, specific methods like Dalkon Shields and Norplant have a checkered safety history, and yet were aggressively marketed to certain populations or as a conditionality imposed by the courts [1, 8, 16]. Sadly, this is not a historical artefact but very present danger. Until the summer of 2021, having an LARC inserted was a condition for at-risk women to access additional social support in the United Kingdom [17]. In the same period, there was a bias towards LARCs over all other methods in post-partum contraceptive counselling in Tanzania [18].

Despite this history of coercing disadvantaged and marginalized women to use LARCs, there have been few safeguards put in place to protect against these perverse practices. For example, a review of WHO's (2016) *Selected Practice Recommendations for Contraceptive Use* [19] and WHO and JHU (2022) *Family Planning: A Global Handbook for Providers* [20] found no active recommendations for safeguarding and ensuring voluntarism in the context of providing LARCs, rather the focus was on safety through assessing users' medical eligibility for

the method and on how best to counsel women. This contrasts with the guidance on sterilization in the same documents that stress the need to ensure the process is voluntary given the history of forced sterilization [20, 21]. We, hence, present this review as a corrective to the silence and inaction around the possibility for coercion in LARC programs.

The evidence on the coercive practices related to LARC has yet to be systematically compiled to support the development of rights-based clinical and programmatic guidance. This evidence gap feeds into and perpetuates the continued uncritical provisioning of LARCs. Therefore, to show the magnitude of the issue, we performed a systematic review of the coercive practices surrounding the provision of LARC. We used a meta-narrative review (MNR) methodology so we could draw on a range of disciplines, specifically law, public health and medicine, and social science. This review is part of a larger research effort by the Global LARC Collaborative (GLC), composed of representatives from Ipas Nepal, FLASCO Argentina, University of California–San Francisco, University of Lancaster, Reproductive Health Uganda and University of Essex, to expand our understanding of coercion in LARC programming globally, and to ensure adequate safeguards are in place to protect against it.

In this paper we set out the methodology we used, the metanarratives we uncovered and outline the types of coercion practices we found. The discussion considers the challenges we faced when examining coercion across disciplines, what this tells us about knowledge practices more generally and what steps are needed to ameliorate these abuses in the future.

## Methods

We conducted a metanarrative review (MNR) of papers documenting human rights violations involving LARCs across three disciplines. A MNR allowed us to draw on heterogeneous, and complex evidence from different disciplines [22, 23]. A MNR recognizes that knowledge about a topic is produced within the particular epistemic tradition of a discipline (e.g., social sciences, biomedicine etc.) and it is necessary to look at the assumptions, values and epistemologies of disciplinary paradigms in order to situate its knowledge [24].

We used WHO's (2014) human rights framework based on agreed standards and definitions to identify violations and abuses, which is commensurate with legal accountability apparati and potential means of remedy and redress [1]. The World Health Organization's (2014) rights-based approach aims to promote, protect, and fulfil the rights of all individuals to choose whether, when, and how many children to have; and to act on those choices through access to quality sexual and reproductive health information and services, free from discrimination and coercion [1, 25].

### Search strategy

Our MNR search strategy set out to respond to the following questions: 1) what is the evidence on the human rights dimensions of LARCs? 2) what are the findings around intentional acts to limit an individual's sexual and reproductive rights? 3) what are the ways these have been addressed? and 4) what insights can be drawn through comparing the findings? We developed a protocol using the RAMESES standards for MNRs [26] which we registered with PROSPERO (number CRD42021268913).

Drawing on relevant existing reviews and consultations with information scientists, experts, and our advisory group, we developed two sets of search terms composed of synonyms for (1) LARCs, and (2) human rights. Between May and July 2021, and updated in May 2022, we searched nine (9) discipline specific databases namely CINHAL, MEDLINE Complete, Academic Search Ultimate, PsycINFO, SocIndex Web of Science, Embase, PubMed, and Lexis-Nexis. In line with a Cochrane Collaboration highly sensitive search strategy, we assessed

**Table 1. Inclusion and exclusion criteria.**

| Inclusion | Exclusion |
|---|---|
| Papers that describe human rights in the context of LARC information and services (e.g., whether by service providers, or through a policy or program) | Papers that relate to intimate partners or personal relationships (e.g., family members, friends) |
| | Papers that relate to women being denied or encouraged to use LARCs based on their clinical eligibility. |
| Any contraception users | |
| Published in English and Spanish language | Published in any language other than English or Spanish |
| Published since 1970 | Published before 1970 |
| Any country | N/A |
| Peer reviewed full text articles | Opinion papers, editorials, dissertations and theses. |

whether our search retrieved the papers that we would expect to be included in the search results.

## Inclusion/Exclusion criteria

We had two phases of inclusion and exclusion. In the first phase, we reviewed the title and abstracts of all the papers against the inclusion and exclusion criteria outlined in Table 1.

In the second phase, we grouped the papers from the first phase into those that could be classified as an unintentional practice and actions (e.g., omissions) and those that could be classed as intentional practices to shape women's decision-making around LARCS (e.g., commissions), see Box 1.

As this review will go on to show, what is a commission and omission is often unclear, and varies by disciplinary paradigm. Therefore, we also created a third category: 'red flags'. Following Karen Hardee et.al (2014), we define 'red flags' as acts or practices that might be hard to label as intentional, but nonetheless represent a normalized practice that is known to amount to violations or abuses based on age, race, socioeconomics, and disadvantaged populations broadly [27]. For example, paying healthcare providers in low-income areas on the basis that they reach a set quota of LARCs can inscribe a political or ideological agenda that certain women and families *should* have less children and can be seen as a 'red flag'. Only papers that presented commissions (e.g., intentional acts to influence a person's reproductive decision) or 'red flags' that could amount to human rights violations (i.e., by the state) or abuses (i.e., by

## Box 1. Key definitions.

Commissions: acts or behaviours that intentionally set out to infringe the human rights of specific groups of people.

Omissions: failures to act, often unintentionally, where there is a duty of care and there is an unintentional failure to fulfil legal duties that results in infringing the human rights of specific groups of people.

Red Flags: violations that might be hard to label intentional, but nonetheless represent a normalized form of practice that is known to create abuses often according to age, race, socioeconomics, and marginalized populations broadly.

non-state actors) were included in this meta-narrative review. We included papers in which women were coerced to use LARCs and those cases where women were forcibly denied access to LARCs.

In the first phase of inclusion and exclusion, a total of 13,134 titles and abstracts were found, and 456 duplicates were removed. Two teams of three people (RDS, VB, RE, LB, SP, MM) dual screened the 12,678 unique records. We selected 684 records for a further round of review by three people (RDS, VB, RE) and from this, 341 records were selected against narrower criteria. At this point, the papers were grouped (by RDS and VB) into two categories: (1) those that exclusively documented unintentional actions, and (2) those that documented intentional practices that tried to unduly influence people's reproductive decisions (e.g., coercion). There were 236 papers that documented omissions underwent a separate review that is not presented here. For this paper, a total of 107 papers underwent full paper review, 14 papers were excluded, one paper could not be accessed, and a total of 92 papers were included (see Fig 1).

## Data extraction and synthesis

Three members of the research team (RDS, VB, and RE) developed data extraction categories that were applied to the included papers. Following from this, one author (RDS) conducted the data extraction of the following information: the year of publication, the author(s), country of study, country income level, study purpose, study design, population, sample size, LARC type, results, limitations, commissions, red flags, authors' discipline(s), theoretical framework (s), historicity, human rights domains used, recommendations, and quality appraisal. To classify authors by discipline, we first consulted author biographies in papers for their departmental affiliations. However, when this information was not available, we assigned papers to the appropriate discipline based upon the methods, type(s) of data presented, and method of data analysis.

To synthesize the material, the authors (RDS and VB) assessed the epistemic traditions used in the papers and this informed the metanarrative categories. Each metanarrative represents a body of knowledge that has a shared paradigmatic approach to documenting and assessing coercion in LARC programs and is not specific to a particular discipline (McDermott et al. 2021). This more paradigmatic approach to grouping allows us more interdisciplinarity than does organizing the metanarratives by discipline. Once grouped, we asked three questions of each metanarrative: how has each tradition conceptualized the topic? what theoretical approaches and methods were used? and what are the main empirical findings?

## Quality assessment

To assess the quality of the papers and the risk of bias, we used the Walsh and Down (2002) [28] to assess qualitative papers and the Higgins and Green (2011) Cochrane Risk of bias tool for quantitative papers [29]. Most papers had a low risk of bias.

## Results

We first present the results from our search and describe the characteristics of the included papers. We then present each of the three metanarratives and then we turn to the types of coercion.

## Characteristics of the papers

Ninety-two papers explicitly documented human rights commissions and/or 'red flags' in LARC programs. Based on our definitions, thirty-nine (39) papers exclusively documented

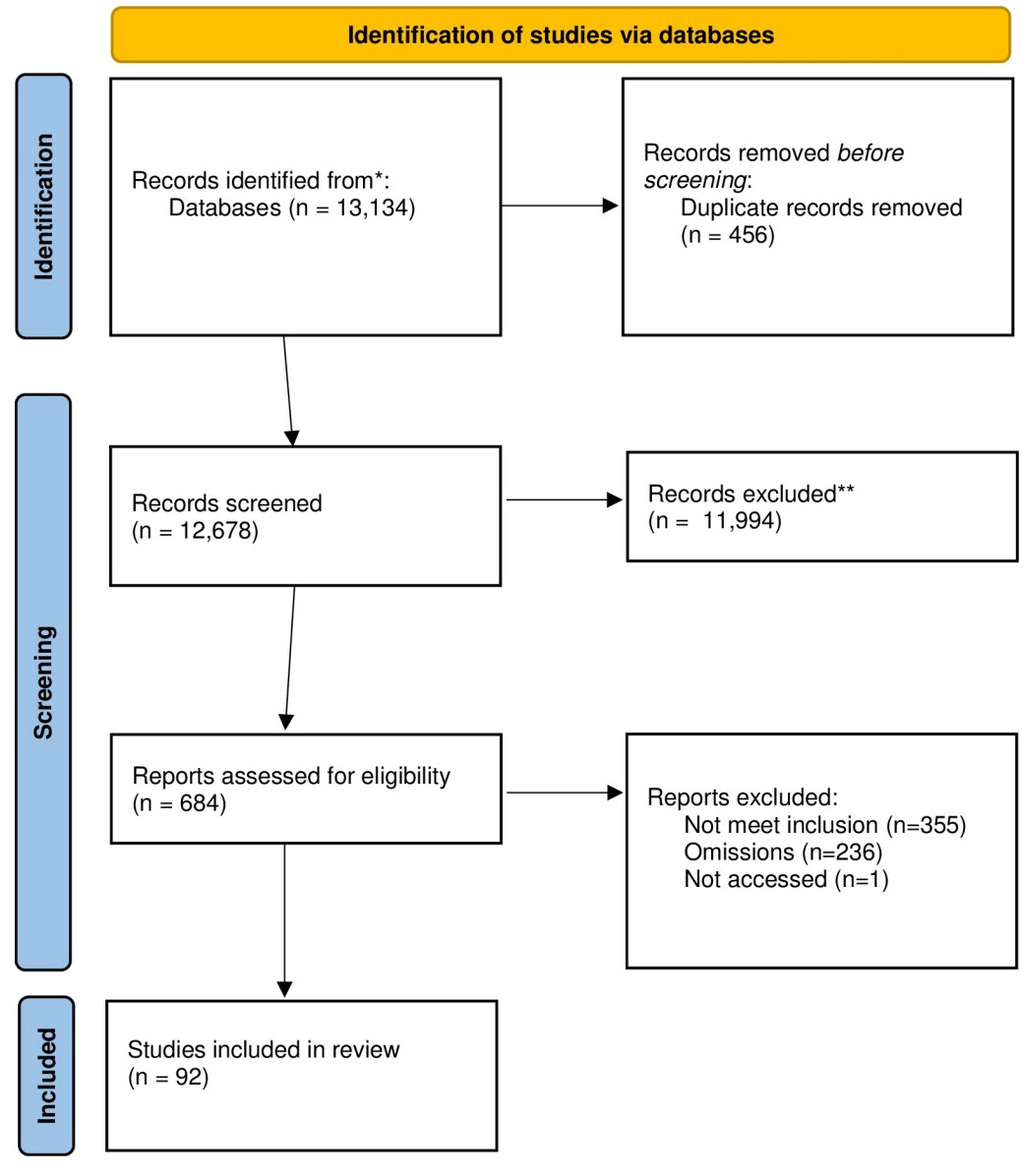

**Fig 1. PRISMA file.**

commissions, 27 papers record commission and red flags, and 26 papers report red flags only. Most of these papers focused on the United States (n = 61). Ten of the studies were in Western European settings, including studies in the UK (n = 6), USA and UK (n = 1), Europe and North America (n = 1), Finland (n = 1), and Sweden and Finland (n = 1). Only 22 papers related to settings outside of the European or North American context and included India (n = 5), Brazil (n = 2), New Zealand (n = 1), South Africa (n = 1), Taiwan (n = 1), China (n = 1), Mexico (n = 1), Puerto Rico (n = 1), New Zealand (n = 1) and Ghana (n = 1). The remaining papers conducted comparative analyses between countries or globally, including a comparison between South Africa, Kenya, Swaziland, and Zambia (n = 1), USA and Bangladesh (n = 1), Indonesia and Jamaica (n = 1), a global comparison of India (n = 1), a global comparison of the USA (n = 1), and a global analysis (n = 1). One paper documented coercion

in LARC programs in Mexico by studying the patients presenting at clinics in the USA. Most papers that performed research in Europe and or North America had representation by authors based in those countries, however, the papers that conducted research outside of Europe and/or North America did not have (significant) representation of authors based in countries covered in the paper.

Many papers focused on specific LARCs such as Norplant (n = 19), Inter-Uterine Device (IUDs) (n = 10), Depo-provera (DP) (n = 9), DPMA (n = 1), and Dalkon Shield (n = 1); multiple LARCs including IUD and oral contraceptives (n = 1), IUDs, pills, and condoms (n = 1), Norplant + Net-EN (n = 1), DPMA + LNG subdermal implant + Copper IUD (n = 1), and Copper IUD + tubal ligation + birth control (n = 1); and some papers used a broader category, such as "LARCs" (n = 30), contraceptives (n = 7), family planning (n = 5), postpartum LARCs (n = 2), reproductive health services (n = 1), contraceptives with a focus on IUDs (n = 1), and hormonal contraceptives with a focus on DP (n = 1).

Four studies were published between 1982–1989, 26 papers were published between 1990–1999, 10 papers were published between 2000–2009, and 52 papers were published between 2010–2021. Papers tended to reflect the then current controversies. For example, papers that were published in the late 20th century tended to concentrate on debates surrounding Norplant, DPMA, and DP. Whereas papers published in the early 21st century centred on LARCs as general category within larger health and political systems. Similarly, different research paradigms have been at the fore of these debates at different times. Law papers were more common in the late 20th century, whereas public health and medicine were more frequent at the end of the 20th and into the early 21st century, and social science papers were most numerous in the early 21st century.

## Overview of the metanarratives

We identified three metanarratives: law, public health and medicine, and the social sciences (see Table 2 for included papers by metanarrative).

Table 3 outlines the characteristics of associated with each of the metanarratives.

## Legal metanarrative

The legal metanarrative around the coercive practices in LARC programming predominantly relates to the examination of specific legal cases in the United States in the late 20th century. Driven by lawyers and legal scholars in law journals, the legal metanarrative appraises, and debates specific legal conundrums triggered by the courts making LARC use a conditionality for post-incarceration probation, the incarceration and/or rehabilitation of male sex offenders, and other ethically ambiguous legal situations.

The American Bar Association (1993) held the view that "any appearance that the woman has 'consented' to the implantation as a condition of probation is strictly illusory. When the alternative to the implantation is prison, the defendant's 'consent' has been coerced [30]." Voluntary consent to Norplant is not possible "in the coercive setting of a criminal trial" [31]. Beyond legal theory, some authors portrayed the courts forcing use of Norplant as coercively "renting a low-income woman's womb" to "further social policy goals. . .among certain targeted groups" that would reproduce social inequalities [16, 32]. The mandatory use of Depo-Provera (DP) for male sex offenders as a form of probation from prison or as a form of 'voluntary' rehabilitative treatment within prison was another legal dilemma related to LARCs. In cases of DP on men, as compared to Norplant on women, the parallels with castration made a more direct argument for violating human rights.

**Table 2. Papers per metanarrative.**

| Law (n = 13) | Public Health & Medicine (n = 59) | Social Science (n = 20) |
|---|---|---|
| Demsky, L.S. (1984); Rhodes, A. M. (1991); Persels, J. (1992); Mertus J., Heller S. (1992); Mubaraki, M. (1992); Rutherford, C. (1992); American Bar Association (1993); Splitz, S. (1993); Albiston, C. (1994); Vance, J. L. (1994); Young, M. E. (1995); Harris, J. G. (1996); Baker, L. (2001). | Savage, W. (1982); Satia, J. K., Maru, R. M. (1986); Melella, J. T. et al. (1989); Nursing Standard (1990); Parsons, C. D. (1990); American Medical Association (1992); Srinivas, K. R., Kanakamala, K. (1992); Egan, T. M. et al. (1993); Moseley, C., Beard, M. T. (1994); Gehlert, S., Lickey, S. (1995); Thompson, M. S. (1996); Frank, M. L., DiMaria, C. (1997); Kirsch, J. D. (1999); Stanback, J., Twum Baah, K. A. (2001); Scott, J.A., Campos-Outcalt, D. (2005); Pelopida, T. M. et al. (2006); Anonymou (2008); McCarthy M. (2009); Volscho, T. W. (2011); Heil, S. H., Higgins, S. T. (2012); White, K., et al. (2015); Gillam, M. L. (2015); Vanthuyne, A., et al. (2015); Burns, B., Grindlay, K., Dennis, A., (2015); Foster, D.G., Barar, R., Gould, H., et al. (2015); Dasari, M., Borrero, S., Akers, AY., et al. (2016); Stevenson, A., et al. (2016); Woo, J. C., et al.(2016); Swartz, M. et al. (2016); Roberts, L., Kaplan, D. (2016); Gubrium, A. C. et al. (2016); Guiahi, M., et al. (2017); Moniz, M.H., Spector-Bagdady, K., Heisler, M., et al. (2017); American College of Obstetricians and Gynecologists (2017); Zeal, C., Higgins, J.A., Newton, S.R. (2018); Brandi, K., Woodhams, E., White, K.O., Mehta, P.K. (2018); Behmer Hansen, R. T., Arora K. S. (2018); Guiahi, M. (2019); Meiera, S., et al. (2019); Newman, K. (2019); Brandi, K., Fuentes, L. (2020); Grzanka, P. R., Schuch, E. (2020); Ma, R. et al. (2020); Ruchman, S. G. et al. (2020); Smith, E., et al. (2020); Skracic, I. (2020); Sznajder, K., Carvajal, DN., Sufrin, C. (2020); Tomar, S., Dehingia, N., Dey, AK., et al. (2020); Biggs, M.A., Tome, L., Mays, A., et al. (2020); McCloskey, L.A., Hitchcock, S., Eloff, I., et al. (2020); Romero, L., Mendoza, Z., Hurst, S., et al. (2020); no author (2020); Senderowicz, L., Higgins, J. (2021); Bryson, A., Koyama, A., Hassan, A. (2021); Cannon, R., White, K., Seifert, B., et al. (2021); Thompson, K., Kirschner, J.H., Irwin, S., et al. (2021); Hill, A.L., Zachor, H., Miller, E., et al. (2021); Relias Media (2021); Senderowicz, L., (2019); Charron et al. (2022) | Nelson, H. L., Nelson, J. L. (1995); Ollila, E., Hemminki, E. (1996); Ollila, E., Hemminki, E. (1997); Van Hollen, C. (1998); Mills, C. (1999); Ollila, E., Hemminki, E. (2000); Takeshita, C. (2004); Volscho, T. W. (2008); Marshall, C., et al. (2009); Welch, M. K. (2010); Takeshita, C. (2010); Lin, Y. D. (2011); Takeshita, C. (2015); Cristina de Lima Pimentel., et al. (2017); Wilson, K. (2018); Mann, E. S., Grzanka, P. R. (2018); Winters, D. J., McLaughlin, A. R. (2019); Sathyamala, C. (2019);<br>Brandao, E.R., Cabral, C.D. (2021), Morison (2022) |

The boundaries of the legal metanarrative become apparent in more ethically ambiguous legal cases where there is no clear precedence [33, 34]. Such ethically ambiguous terrains consist of the status of the foetus and protection of existing children. For example, in Young's

**Table 3. Characteristics of the metanarratives.**

| Law | Public Health and Medicine | Social Science |
|---|---|---|
| The majority published in late 20th century.<br>• There is a focus on coercive use of LARCs, predominantly emerging from specific legal cases in the USA.<br>• Presents an outline of debates and legal puzzles.<br>• References a codified set of norms as outlined in human, rights and constitutional law.<br>• The legal person is the object of analysis with less focus on the systems and structures surrounding them.<br>• Targets marginalized, disadvantaged and excluded population(s). | Published across the late 20th and early 21st century.<br>• A focus on the instruments of the health system that create conditions for coercion.<br>• Presents an understanding of situations and tries to change them.<br>• References norms and standards codified and cascaded in laws, policies and regulations.<br>• The object of analysis is the system (or experience of the system) and less focus on the individual.<br>• Targets marginalized, disadvantaged and excluded population(s). | • The majority published more recently in the last 15 years.<br>• A focus on thick descriptions of the more dispersed and indirect forms of coercive practices that are contextual and relational.<br>• Presents accounts that are descriptive and evaluative in nature.<br>• Draws on the precepts of reproductive justice that are not regulations or codes, but principles.<br>• The subject of analysis is the intersection between structures and individuals, with specific attention to intersections between them.<br>• Targets marginalized, disadvantaged and excluded population(s). |

(1995) examination of "conflicts between the state's responsibility in protecting the interests of future children and the mother's interest in control over her own reproductive life," Young argues that "we must see reproductive technologies as confirmations of dominant social conclusions about gendered and ethnic reproductive behavior and norms [35]." These cases have seemingly little judicial precedence in terms of case law; yet there is significant precedence in how intersecting ideologies about race, socio-economic status and fit parenting become (re) inscribed into rationalities about reproductive practices. However, in the absence of judicial precedence, these moralizing undertones are masked in the unsettled legal debate.

The language of human rights was commonly used as was the focus on the legal person. The legal metanarrative is premised on the idea of legal personhood in which an individual subject is capable of being designated rights, protections, duties and responsibilities by the law. The focus on individuals in the legal metanarrative worked to obscure the broader ideologies and structural forces at work, those that, for example, make possible and legitimize a poor, women of color being imagined as an 'unfit parent.'

## Public health and medicine narrative

In contrast to the focus on the individual, the public health and medicine metanarrative focuses on how the systems, structures and processes of the health system can exert influence on an individual's decisions. As a more systems approach in which the health system itself as an all-encompassing rational instrument that (singularly) informs and organizes practices and institutions, this metanarrative focuses on what makes the system work, how it can be influenced, and if there are system deficits, how they can be resolved. These papers began to appear in the late 20th century following legal debates and continued into the 21st century alongside developments of reproductive rights and justice.

Interventions e.g., strategic actions to alter a result or course of events, are a key feature of this metanarrative. Interventions could be those that attempt to directly influence people's decisions using incentives to encourage a certain behaviour, which are more easily identified as coercive as they distort consent and decision-making. Interventions can also apply at a systems level, such as changing the commodity supply chains or excluding training in LARC removal, that are less likely to be classified as intentional but rather as 'red flags.' Yet both types of intervention have the same effects on individuals. The description of interventions tended not to be situated within the wider social, political and economic contexts in which they occurred.

System wide changes were also seen to be triggered by laws, policies and regulations. Policies changes, such as cuts in funding to specific services, the prohibition of specific services on religious grounds, or setting age restrictions on certain contraceptive methods, become coercive when they limit the services available and thus restrict people's choices. Apart from funding cuts—whereby we felt it was reasonable to assume that cutting existing funding was intentional—we usually classified policies as 'red flags' because it was unclear whether the outcomes were explicit attempts to induce a particular behaviour. Like interventions, laws, policies, and regulations were seen to have detrimental effects for individuals, yet the wider contexts and ideologies underlying these policies were un(der)analysed.

Understanding the contraceptive users' interaction with the system was another feature of this metanarrative. Users' interactions with systems and structures were often limited to clinical consultation, with a particular focus on patient-physician interactions and directive counselling. Zooming in on the clinical consultation may highlight the perverse effects of tiered counselling but it often neglected the structural factors beyond the clinic that facilitate and even encourage such interactions. Unlike the focus on interventions and policies, the focus on

users' interactions with the health system did identify the specific actors and spaces where coercion can occur, how it occurs and how different actors navigate these circumstances.

This metanarrative cuts across disciplines, employs a variety of quantitative and qualitative methods, and traverses a variety of conceptual frameworks and subjects of study. Across all papers, historical and conceptual information was lacking, and did not play a significant part in the analysis. At most, these historical and conceptual frames helped authors to establish research questions, but these frameworks were rarely used to explain their findings. By taking the health system, its interventions and polices as the object (and often sole subject) of study, the public health and medicine metanarrative rarely accounted for the broader moral and political rationalities that produce and mediate coercive practices.

## Social science metanarrative

The social science metanarrative did not lean towards a systems or individual approach, but rather it focused on the interactions between structure, systems and individuals and the logics and rationalities that link them. Through thick empirical descriptions, extensive use of critical social theory, and capturing the intersections of power, this metanarrative demonstrates how certain rationalities cut across bodies, systems, policies and programs, and global political economies. These papers overwhelmingly appeared in the early 21st century, notably at a time when the critical discourse on reproductive justice became widespread.

The thick descriptions that are characteristics of social science relay detailed explanations of situations, and as such the social science metanarrative draws out the historical, political, social and economic contexts of coercive practices to make sense of their empirical material. This strong focus on context brings a historical lens that highlights the continuities between past and present, and how, while forms of coercion may have changed, the underlying ideologies that justify their practice often remain consistent. This metanarrative also looks at the broader contemporary social contexts of coercive practices surrounding LARCs; this includes recognizing the various actors and subject positions and acknowledging the many institutions—whether religious, political, commercial—that together produce the conditions of stratified reproduction—the valuing of certain groups' reproductive rights over others [36].

By juxtaposing empirical material with its wider contexts, the forms that stratify reproduction become more apparent. Specifically, context shows how actors' actions help to constitute systemic ideological power structures, and conversely how institutions articulate these ideological power structures in ways that seemingly normalize coercion [37]. By using context to elucidate this process, the social science metanarrative shows how stratified reproduction is both the ideology driving coercive practices as well as its outcome. Therefore, this metanarrative shows how the system itself performs acts of commission by encouraging the reproduction of a 'certain people' categorized in racial or economic terms [38].

There is no normative or binding frame underpinning this metanarrative as there is in the legal metanarrative, but there are frequent references to the precepts of reproductive justice. Reproductive justice is an intersectional theoretical framework for understanding reproductive politics that encompasses the right to have a child, the right to not have a child, and the right to parent a child [39]. Connecting to reproductive justice in this metanarrative provides an analytical framework whereby "eugenic practices are elucidated as situated at the complex intersections of privilege and oppression" [40, 41]. Reproductive justice allows for powerful and nuanced descriptions, yet there is less emphasis on accountability, and on finding remedies and solutions associated with the legal or health-oriented metanarratives.

## Coercive practices surrounding Long-Acting Reversible Contraceptive (LARC) methods

Across the three metanarratives, there is one clear conclusion: all instances of coercive practices in LARC programs target marginalized, disadvantaged and excluded population(s). Regardless of the metanarrative, coercive practices are always related to a specific population of women: indigenous Adivasi and Dalit women [41–43], African American and First Nation women [33, 43, 44], low-income women [45, 46], young people [47, 48], substance users [49–52], marginalized women in the global south [40, 53–58], women in the carceral system [32, 46, 59–61], women receiving welfare [34], women with psychiatric conditions and intellectual disabilities [62], and women in refugee settings [62]. This clear association of LARCs with the coercion of certain groups of people demands our attention.

## Continuum of coercive practices

Coercive practices, as Senderowicz (2019) states, '*sit on a spectrum and need not involve overt force or violence but can also result from more quotidian limits to free, full and informed choice'* [63]. A similar spectrum correlates with LARCs, Fig 2 outlines a continuum of the reported cases we found and recognizes the range of practices and actions that unduly influence people's reproductive decisions. This can encompass both more direct and overt actions through to much more implicit actions of morally normalized practices that coalesce to restrict people's choices. The more direct and overt coercive practices, such as inserting an IUD post-partum without seeking consent, clearly passes the threshold of established legal code that we tend to associate with the violation of human rights. Whereas giving biased information about a method that emphasizes its effectiveness while playing down or not mentioning its adverse effect, are instances in which wider structural conditions and systems work to constrain an individual's choice. While this may not meet the threshold of a human rights abuse it is typical of normalized, systematic forms of practice that distort the process of consent and result in forcing a person to do something against their will.

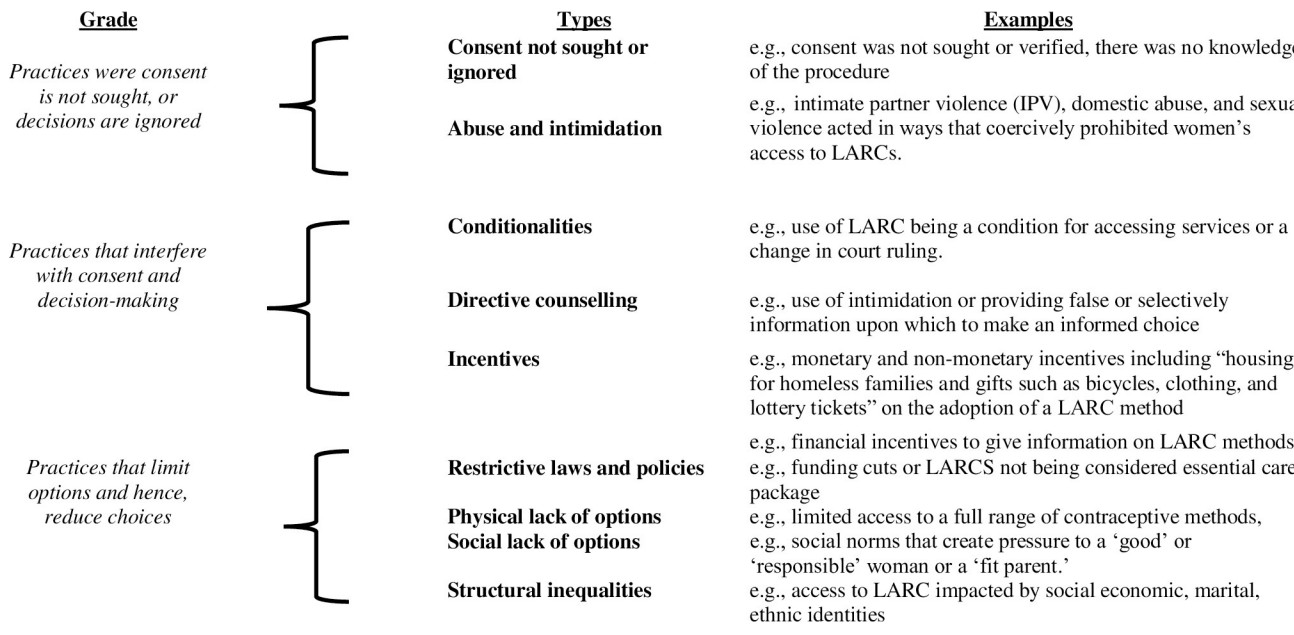

| Grade | Types | Examples |
|---|---|---|
| *Practices were consent is not sought, or decisions are ignored* | **Consent not sought or ignored** | e.g., consent was not sought or verified, there was no knowledge of the procedure |
| | **Abuse and intimidation** | e.g., intimate partner violence (IPV), domestic abuse, and sexual violence acted in ways that coercively prohibited women's access to LARCs. |
| *Practices that interfere with consent and decision-making* | **Conditionalities** | e.g., use of LARC being a condition for accessing services or a change in court ruling. |
| | **Directive counselling** | e.g., use of intimidation or providing false or selectively information upon which to make an informed choice |
| | **Incentives** | e.g., monetary and non-monetary incentives including "housing for homeless families and gifts such as bicycles, clothing, and lottery tickets" on the adoption of a LARC method |
| *Practices that limit options and hence, reduce choices* | **Restrictive laws and policies** | e.g., financial incentives to give information on LARC methods e.g., funding cuts or LARCS not being considered essential care package |
| | **Physical lack of options** **Social lack of options** | e.g., limited access to a full range of contraceptive methods, e.g., social norms that create pressure to a 'good' or 'responsible' woman or a 'fit parent.' |
| | **Structural inequalities** | e.g., access to LARC impacted by social economic, marital, ethnic identities |

**Fig 2. Continuum of coercive practices related to LARCs.**

Therefore, these forms of coercions are not mutually exclusive but are rather co-constituting: the more normalized forms of institutional practice can work to empower more overt forms of coercions that are enacted. With this reflection, we propose a continuum as a tool that gradates the various of forms of coercion found and attempts to put them in relation to each other. However, we caution that categorizing these forms of coercion along a continuum can also risk losing nuance; specifically, regarding the time periods when specific coercive practices were common, and the forms that coercive practices may take in those times. Direct coercions were more likely to be documented since the 20[th] century, while indirect coercions were more likely to be documented in the past two decades as discourses of reproductive justice became increasingly well-known. However, in some parts of the continuum there are more recent examples of direct coercion and older examples of indirect coercion. In some cases, this also meant that at times the coerced use of contraception sat alongside limited access to contraception which also restricted self-determination.

Here, we illustrate the degrees along the continuum by documenting specific coercive practices that constrain women's agency to consent to using LARCs.

There are coercive practices where consent is not sought, or women's decisions are ignored, and these pass the legal threshold for a human rights violation or abuse.

**No consent.**   Egregious forms of coercion were the instances in which women were provided a long-acting method without their knowledge and/or their consent was not sought. This includes cases of IUDs being inserted post-partum without women's agreement [42, 55, 64]. Campos-Outcalt (2005) found that about "27% of women presenting at [their] clinic [which] had received insertion in Mexico" had the IUDs inserted without their knowledge [55]. Similar practices have been seen in India, where Van Hollen (1998) found women receiving IUDs in Tamil Nadu, often without consent, in the context of the All-India Post-Partum Program as part of a 2-child policy [54]. Van Hollen documents how, because healthcare practitioners "lacked an outreach system through which they could trace the women and try to motivate them after they" gave birth in the hospital, healthcare practitioners "tended to view them [women] as 'moving targets'—difficult to reach" to accomplish their population control goals leading to IUD administration without consent. Similarly, contraceptives, specifically DP, have been administered without consent for women confined to institutions [65].

**Abuse.**   There were several examples of how intimate partner violence (IPV), domestic abuse, and sexual violence acted in ways that coercively prohibited women's access to LARCs. For example, when a woman tried to access LARC it led to intimate partner violence [66–68]. This type of coercion had further harmful effects such as increasing women's odds of contracting an HIV infection [69]. Examples of this type of coercion, often from the metanarrative of public health and medicine, were unique in their ability to connect coercion to its source and consequences of further human rights violations.

There are a range of coercive practices that interfere with and distort the process of giving consent.

**Conditionalities.**   In the US context, there were many examples of court rulings on the mandatory use of Norplant as a condition of probation [40, 46, 70, 71]. In 1993, the American Bar Association (ABA) cautioned that the requirement of Norplant as "a condition of probation or a condition of welfare benefits" violates constitutional rights. [30] Any notion that a woman has 'consented' to Norplant as a condition of probation is 'illusory' when the alternative is imprisonment. A review of imprisoned women in the USA showed how women are coerced into using LARCs in order to leave prison on probation, and how once having left prison these women often lack access to healthcare providers to remove them [40]. How Norplant has been used by courts and proposed legislation in violations of constitutional rights

relates to broader social (population) control—particularly forcing poor, women of color to use LARCs [40, 46, 70, 71].

**Biased and directive counselling.** Consent can be distorted through interactions with healthcare providers or counsellors. This could include healthcare providers sharing biased information based on what they think about a patient or a contraceptive method [72]. For instance, implying that certain methods are only appropriate for women who have already had a child, using scare tactics about the implications of having another child [18] or counselling at vulnerable moments, such as post-partum or abortion [73]. Bertrand et al. (1995) defined this as "the practice of favoring some methods and discouraging others in the absence of a sound medical rationale, as well as failing to ascertain and to respect the client's preference" [74]. In the UK, there were instances in which "inadequate information provided to women about the side-effects of" controversial LARCs manipulate their choices [44], and in Indonesia and Jamaica women were "given little information upon which to make an informed choice" about DP [75]. Interviews with physicians in 46 family planning facilities in Ghana found physicians imposed their own morals on what are acceptable forms of LARC provision [76]. Physicians may also have a bias for particular groups, such as recommending longer acting methods to youth and homeless populations [66, 77]. Another way in which healthcare providers distort reproductive decision-making is through directive counselling, in which a provider selectively shares information–for example only talking about the advantages of a certain method and downplaying its side-effects [52]. The ACOG (2022) describes directive counselling as instances where counselling "may be subject to undue influence, such as a counsellor's personal biases (implicit or explicit), pressure or coercion from a counselor or partner, or even the ideology of the institution at which someone is seeking contraceptive access" [78].

**Incentives.** Incentives are often used to encourage particular behaviours, this could be through rewarding a behaviour with some kind of bonus, reward, or payment [79]. In this review, we found incentives aimed at patients and those aimed at providers. For example, in 2019, the Project Prevention Program in the US offered USD$300 to substance users to be sterilized or use contraceptives. This program targeted "people who are living in poverty" who may need immediate sources of income but who may also make choices that they later regret [51]. Moreover, incentive programs for substance users to obtain welfare benefits in the USA "demonstrate the pervasiveness of reproductive control in criminal justice and criminalizing social welfare systems," particularly for low-income, women of color [40]. Satia and Maru (1986) documented the use of both monetary and non-monetary incentives including "housing for homeless families and gifts such as bicycles, clothing, and lottery tickets (137)" in India [80].

Incentives are also used to encourage health providers to promote LARCs among their patients. Ma et al. (2020) described how health care practitioners in the UK: "were given financial incentives to give information on LARC methods to over 90% of women coming for contraceptive devices" [81]. The study reviewed over 3 million health records between 2004 and 2014 and found that these financial incentives were "associated with more LARC prescriptions and reductions in abortions . . . particularly in younger women aged 20–24 and those from poorer backgrounds (3)."

There are also coercive practices that lead to conditions that limit the options available and hence, reduce choice and prevent women from enacting their choices.

**Laws and policies.** Policies can be coercive when they work to influence people's choices directly or by restricting the options available to them. Some cases caution against the introduction of policies in certain settings as there are historical precedents for coercion. For example, in India, amidst proposals to introduce Norplant into publicly funded family planning programs that historically targeted women from "lower strata of the society", Srinivas and

Kanakamala (1992) cautioned how Norplant was "already being used by US courts as a means of punishing women and controlling their fertility" [82].

In addition to coercing the use of contraception, policies can also limit access to contraception by restricting access to services and consequently what options are open to people. For instance, in 2011 the state of Texas cut funding to family planning programs that reduced the number of clinics and appointments. Hence, women were unable to find providers and this disproportionately affected socio-economically disadvantaged groups [83, 84] and young people [85]. Similar cuts in Arkansas found that "choices about whether and when to have children have been inextricably linked to women's economic situation and how birth control initiatives were often justified as instruments of control rather than individual self-determination (220)" [57]. Such funding cuts are more likely to narrow contraceptive choices for poor, oppressed, and marginalized women.

Institutional policies can work to restrict services, for example clinics owned and funded by the Catholic Church in the USA [86, 87]. Often the willingness of such clinics to offer services is not only based upon institutional policies but also on the social and cultural norms. Guiahi et al. (2017) argue that "patient autonomy is violated not only because they are unable to receive care they want or need, but because patients are not given the opportunity to avoid conflicts in care by being informed ahead of time about LARC restrictions (6)" [86]. More recently, the policy responses to the COVID-19 pandemic have restricted access to LARCs. Senderowicz and Higgins (2021) documented how LARCs were deemed as non-essential during the COVID-19 pandemic in the USA, whereby institutions have intentionally reduced services and subsequently restricted reproductive autonomy[56]. Another common service restriction relates to age. Policies whereby adolescents must have parental consent to access LARCs poses a 'red flag' for adolescents' self-determination and bodily autonomy and reinforces a "stigma regarding adolescent sexuality" [48].

**Restricted options.** Wider structural factors can distort people's reproductive decision-making through limiting the options that are available to them. Options could be material (e.g., a limited range of options offered) or it could be restrictions imposed by what is considered socially permissible and acceptable behaviour. Women who are dependent on public sector services are particularly vulnerable to this form of coercion. For example, Wilson (2018) analyzed how Indian family planning programs that targeted "poor, Adivasi and Dalit women," only provided access to sterilization for women who were actively seeking contraception as there were no other methods available. For Wilson, "this is an example of the way in which coercion is also often embedded in situations that are framed in terms of the exercise of 'choice'" [41, 88].

Social expectations about behaviours also work to restrict options. For example, low-income, incarcerated, women of color on probation in the USA experienced greater governmental intrusion in their private lives because they rely upon (often public) welfare services for economic support and access to public health facilities that provide healthcare [32]. This reliance upon publicly funded institutions subjects them to a heightened scrutiny of their parenting, fuelled by a racist assumption about women of color's 'inability' in child-rearing behavior and restricts their access to LARCs by indirectly coercing them into the 'right' choice. Powerful social expectations about being a 'good woman,' 'a fit parent' and acting responsibly pressure women to use methods that they may not use otherwise in order to meet these ideals [33, 54]. Similarly, contexts of intimate partner violence, women who need to conceal the contraceptive use are often restricted to using LARCs. In instances of intimate partner violence, Skracic (2020) examined how this made LARC the only possible method as it was less detectable, whereas other methods may be partner-dependent and runs the risk reproductive coercion [89].

Finally, the public pressure from the media can encourage women to select certain options over others [90]. For example, the publicly funded media promotion of Norplant in the UK

targeted at-risk women [90]. Further, Mann and Grzanka (2018) illustrated how media advertisements create "a social imperative for subjects to cultivate responsible sexual identities and practices—whereby responsibility is defined in terms of independence from social welfare systems and by adherence to traditional notions of monogamous heterosexuality (i.e., heteronormativity) that reflect the dominant interests of the advanced capitalist state (5)" [91]. For Mann and Grzanka, "the absence of alternatives to LARC communicates that choosing a different contraceptive method, or choosing not to contracept at all, would be a bad—and implicitly irresponsible—choice . . . [this] comes to turn one specific 'choice' into an imperative" [91].

**Structural inequalities.** Structural and historical oppressions shape reproductive decision making in the present. Several papers point out how structural conditions (e.g., criminal justice system, welfare and/or racism) act as determinants of coercive practices related to LARCs. For instance, the AMA (1992) directly cautioned against the use of Norplant as a condition of prison probation and as a financial incentive for women on welfare [92]. Volscho (2011) retrospectively analyzed 41,708 respondents in a CDC survey and found that "African American and American Indian women are more likely to use Depo-provera than European American women." Volscho offers the concept of 'sterilization racism' to claim that even though these women are not subject to permanent sterilization, the long-term effects of DP, particularly if used repeatedly, effectively reduce women's reproductive years [43].

In South Carolina, Smith et al. (2020) illustrate how "women of color understood their reproduction within a larger historical context . . . women of color specifically noted historical oppressions and subsequent impacts on their reproductive health (17)" [93]' Historical trauma manifest in the present in that "restrictive reproductive policies due to traditional, conservative power structures, particularly in South Carolina, negatively impact women's reproductive health." Whereas Grzanka and Schuch (2020) outlined how the experiences of young women, 18–24 years old and majority white, negotiating LARCs were targeted in the USA [45]. In this case, LARC is promoted uncritically, and the historical and reproductive oppression experienced by women of color is replaced by a focus on individual behaviours. Here the concept of "reproductive anxiety" and "conditional agency" can help describe these experiences as the terms reflect how LARC promotion is often framed through a discourse of risk.

## Discussion

This review presents a critical examination of coercion in the context of LARC programs. LARCs are an important and highly effective contraceptive options available to women globally when promoted and provided within a human rights framework. However, we identified examples of coercion in LARC programs on an international scale that undermine self-determination and bodily autonomy and can amount to human rights violations and abuses. In this discussion, we critically consider these findings as part of wider reflection on reproductive governance, on contraceptive programming and finally we outline the limitations of this work.

In the process of conducting the review, we encountered many instances when the boundaries of our analytical categories began to blur. The first blurring happened when we applied the concepts of commission and omission to classify the papers for inclusion. We had hoped to use the degree of intentionality (e.g., the level of awareness and directedness of actions) that separates the concept of commission from that of omission to group the papers, yet we found that in some papers, it was unclear whether the coercion was intentional or not. We, therefore, created a third category of 'red flags' to reflect instances that might be hard to class as "intentional" but clearly represented a normalized practice of abuse.

Once we had selected the commissions and 'red flags', we then encountered yet another blurring around the types of coercions. Because our analysis traversed disciplines, what was

classed as coercion by one discipline was not always categorized as coercion by another. This difference created a blurring of where and to whom coercion may occur. Some authors argued that coercion took place between individuals (law), other authors argued that coercion took place between the health system and an individual (PH and medicine), and other authors argued coercion took place between ideological structures and populations (social science). The different metanarratives tended to report types of coercion that were more commensurate with their understanding of the world and these different epistemologies tended to fragment our knowledge about coercion.

For example, the law metanarrative focused mainly on conditionalities, as this was the most common site of judicial activity. Public health and medicine metanarrative reported cases of direct coercion, often reported as clinical case studies that came with an implicit impetus to address the underlying systems error, aligning with the public health and medicine's epistemologies of systems approach. Whereas the social science metanarrative tended to focus on practices that lead to conditions that limit the options available. The legal metanarrative was most explicit about human rights violations, whereas, human rights language was rarely used in the public health and social science metanarratives, and consequently there was rarely reference or call for accountability and remedy.

Each metanarrative assigned responsibility for coercion differently. For example, the legal metanarrative was more likely to document an individual's responsibility over systemic or moral dynamics, whereas the public health and medicine metanarrative was more likely to look at how systems were responsible. Social science, however, looked at both individual and systems level complicity, and the relationship between them and, in doing so, made clear the wider ideological structures and political rationalities that link them. There was an active consideration how individual actions were tied to local and national systems of governance and the wider global logics, linking 'intimate governance to world governance' [94].

Recognizing that there are different ways of understanding coercion begs the question: how do these different approaches affect our understanding of coercion more broadly? Each metanarrative gives us a partial view of the workings of coercion. By looking across meta-narratives, we gain a fuller picture that enables us to explore the linkages between seemingly disparate events and gain an understanding of the political rationalities that underpin and connect them. Only in bringing these approaches together can we start to understand the extent and nature of coercion and how it traverses a range of types and sources. Coercion covers violent physical abuse through to the more subtle machinery of exclusion, coercion can be both clearly evidenced but also dispersed into a normalized pattern of abuse, and coercion can be meted out by persons, by systems and by deep-seated ideologies.

The patchwork of knowledge we have on coercion in LARCs programming results in a kind of fragmentation, which itself is a knowledge practice that can inadvertently inhibit our ability to collect accurate information on this topic, and group related phenomena together, and consequently contributing to ignorance and inaction. Scholars of ignorance studies have long argued that not knowing something has its own instrumental value. Strategic ignorance entails shutting out inconvenient information and consciously avoiding knowledge can constrict our vision, thereby, issues are left unseen and their causes unchallenged [95, 96]. As many have illustrated in the strategic ignorance surrounding the tobacco industry and climate change, ignorance is how certain groups deny, justify, or ignore injustices against the less powerful [95].

A conclusive finding of this review is that the coercive practices surrounding LARC programs target specific disadvantaged and marginalized populations; yet, the guidelines supporting the provision of LARC fail to explicitly address this propensity. Our inability to collect accurate information about coercion in LARC programs and its subsequent absence from the

relevant guidance reinforces health, economic, and power disparities in sexual and reproductive health practice. In this review, we can see how certain knowledge practices can work to produce ignorance and how this knowledge practice recasts a morally suspect practice as "defensible and even rational (242)" [97].

In some cases, LARCs have become a standard feature of international and national SRHR programs and by design, these programs intentionally target marginalized communities and therefore, any failures inevitably impact these communities. The relative lack of power of women to give their full and informed consent and defend their rights as they are engaged in these programs, requires that active steps are put in place—additional and specific precautions—to explicitly ensure that programs are NOT instruments of discrimination, nor reinforcing disadvantage and discrimination. A path to achieve this is to uplift the agency of the population(s) being served i.e., by respecting and upholding their rights to active, free, prior and informed decision-making as they—without fear or favour—choose LARCs from among other methods options or none at all. First this requires recognizing and responding to the potential for coercion in LARC programming and ensuring appropriate safeguards are put in place to protect against such abuses in guidelines and standards. This requires further research on what are effective ways to actively identify and address coercion in programming–could we use Client Exit Interviews that are standard part of quality assurance, and/or add specific provisions in contraceptive counselling, and/or developing safe and effective reporting mechanisms? A focus on patient-centred care, defined as care that is respectful of and responsive to an individual's preferences, needs, and values over programmatic or biomedical priorities, may be beneficial [98, 99]. Patient-centred care focuses on high-quality and meaningful patient-centred interpersonal communication through which patients' express their needs and preferences and providers' actively facilitate their preferences. This requires both training and support of health care professionals in patient-centred care and shared decision-making models, and including patient-centred counselling in supervision, facility, and program assessment.

## Limitations

The review was limited to three broad disciplines which may have affected what papers were found and included in this analysis. Another limitation of the review is that several papers, mostly in the social sciences, were not identified with our human rights derived search terms in the title and/or abstract and, therefore, were not included (see for example Morrison and Eagar 2022 [100]; Brian et al. 2020 [101]; Gómez and Wapman 2017 [102]). A notable limitation is the bias towards studies in North America and North-Western Europe. We see this imbalance as a form of epistemic injustice resulting from the structural prejudices that emerge in producing, using, and circulating knowledge [103]. The fact that there is limited research on coercive practices related to LARC in diverse global contexts further contributes to this phenomenon going unnamed or unmarked. The literature we captured may not yet reflect the full impact of the COVID-19 pandemic on coercive practices surrounding LARC programs as such papers may not yet be published. A further limitation is that the way in which coercion has been viewed over the past 50 years has varied. The more recent papers were published the more likely that they may exercise a wider analytical lens when considering what may fall within the scope of coercive practices. This means that while there may be less information about structural factors of coercion in the late 20th century, these did exist and were likely much more prevalent than what this review was able to document.

All the papers focused on the instance of coercion and not on the longer term social and emotional ramifications of coercion on people's lives, recognizing that there are multiple

harms resulting from such abuse [104]. There is little research on women's own narratives of lived experience of coercion and abuse outside the incident itself [104]. The lived experience can provide insights into how coercion functions and the emotional and social ramifications, and how we can better safeguard against coercion.

## Conclusion

All the evidence supports the conclusion that coercive practices surrounding LARC programs target specific disadvantaged, excluded and marginalized population(s). Yet the fragmentation of the existing evidence make it difficult to collect accurate information and engender the necessary safeguards. We present this review to address longstanding silence around coercion in LARC programming and encourage clinical and programmatic guidance to actively safeguard against the propensity to coercion and upholds reproductive rights and autonomy.

## Supporting information

**S1 Checklist. PRISMA checklist for MNR.**
(DOCX)

**S1 File. Table of included papers.**
(DOCX)

## Acknowledgments

We would like to acknowledge the support of the Global LARC and Rights Collaborative, namely Mark Limmer, Rachael Eastham, Kelsey Holt, Peter Ibembe, Lawrence Muhungi, Lhamo Sherpa, Manju Maharjan, Kelsey Holt, Lara Crystal-Ornelas, Raul Mercer, Carlotta Ramirez and Sophie Patterson, who reviewed the protocol, reviewed the search terms and commented on the findings. We are particularly grateful to Rachael Eastham, Louise Brennan, Sophie Patterson, and Manju Maharjan for reviewing titles and abstracts for the search.

## Author Contributions

**Conceptualization:** Victoria Boydell.

**Data curation:** Victoria Boydell, Robert Dean Smith.

**Formal analysis:** Victoria Boydell, Robert Dean Smith.

**Investigation:** Victoria Boydell, Robert Dean Smith.

**Methodology:** Victoria Boydell.

**Project administration:** Victoria Boydell.

**Resources:** Victoria Boydell, Robert Dean Smith.

**Supervision:** Victoria Boydell.

**Validation:** Victoria Boydell, Robert Dean Smith.

**Writing – original draft:** Victoria Boydell, Robert Dean Smith.

**Writing – review & editing:** Victoria Boydell, Robert Dean Smith.

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
