## [Decision Letter · Decision Letter 0]

7 Mar 2023

PGPH-D-23-00102

Hidden in plain sight: A systematic review of coercion and Long-Acting Reversible Contraceptive Methods (LARC).

Dear Dr. Boydell,

Thank you for submitting your manuscript to PLOS Global Public Health. After careful consideration, we feel that it has merit but does not fully meet PLOS Global Public Health’s publication criteria as it currently stands. Therefore, we invite you to submit a revised version of the manuscript that addresses the points raised during the review process.

We look forward to receiving your revised manuscript.

Kind regards,

Ateeb Ahmad Parray, BDS, MSS, MPH

Academic Editor

Journal Requirements:

2. Please send a completed 'Competing Interests' statement, including any COIs declared by your co-authors. If you have no competing interests to declare, please state "The authors have declared that no competing interests exist". Otherwise please declare all competing interests beginning with the statement "I have read the journal's policy and the authors of this manuscript have the following competing interests:"

3. Please amend your detailed Financial Disclosure statement. This is published with the article. It must therefore be completed in full sentences and contain the exact wording you wish to be published.

4. Please provide separate figure files in .tif or .eps format only and remove any figures embedded in your manuscript file. Please also ensure that all files are under our size limit of 10MB.

5. We have noticed that you have uploaded Supporting Information files, but you have not included a list of legends. Please add a full list of legends for your Supporting Information files after the references list.

Additional Editor Comments (if provided):

Reviewers' comments:

Reviewer's Responses to Questions

**Comments to the Author**

1. Does this manuscript meet PLOS Global Public Health’s publication criteria? Is the manuscript technically sound, and do the data support the conclusions? The manuscript must describe methodologically and ethically rigorous research with conclusions that are appropriately drawn based on the data presented.

Reviewer #1: Yes

Reviewer #2: Yes

2. Has the statistical analysis been performed appropriately and rigorously?

Reviewer #1: N/A

Reviewer #2: N/A

3. Have the authors made all data underlying the findings in their manuscript fully available (please refer to the Data Availability Statement at the start of the manuscript PDF file)?

Reviewer #1: Yes

Reviewer #2: No

4. Is the manuscript presented in an intelligible fashion and written in standard English?

Reviewer #1: Yes

Reviewer #2: Yes

5. Review Comments to the Author

Reviewer #1: I believe that the research is sound and the publication is ready. The only hesitation I have is that the data is drawn from publications dating back to the 1980s and 1990s, which may not be directly comparable with writings from the 2000s because the discourse of reproductive choice has changed quite a bit. In some ways the contexts have been lost. Another point I would like to bring up is that "coerced contraception" and the "coerced non-access to contraception" appears in the same section even though these have opposite repercussions. I felt that they should be separated for clarity.

Reviewer #2: The authors have presented a review meta-narrative with title "Hidden in plain sight: A systematic review of coercion and Long-Acting Reversible Contraceptive Methods (LARC)". The study concept is relevant to public health and future policy and programme on contraception as regards LARCs. The methods adopted for the review is standard and novel while the general write-up is clear and addressed the aim of the review.

In general the review is limited by the available evidence which is skewed to the US and Europe leaving a large lacuna with the LMICs. The introduction section should have background review of the dimension (Law, public health and social) to LARC coercion. In addition, The authors needs to provide clear evidence on when contraceptive service provision becomes coercive towards LARC as against when it is based on empirical decisions.

The narratives in the result are generally skewed towards coercion irrespective of underline empirical choice of LARCs in the instances given. The authors need to clearly draw a line when the choice of LARCs becomes empirical option and when it is unduly coercive. This is important for a balanced narration.

The authors discussed mostly the review process rather than discussing the findings of the meta-narratives. For instance, the readers will expect to see implications for different spectrum of the coercion and how they apply to future contraception programming.

6. PLOS authors have the option to publish the peer review history of their article (what does this mean?). If published, this will include your full peer review and any attached files.

**Do you want your identity to be public for this peer review?** For information about this choice, including consent withdrawal, please see our Privacy Policy.

Reviewer #1: No

Reviewer #2: No

<quillbot-extension-portal></quillbot-extension-portal>

---

## [Decision Letter · Decision Letter 1]

22 May 2023

PGPH-D-23-00102R1

Hidden in plain sight: A systematic review of coercion and Long-Acting Reversible Contraceptive Methods (LARC).

Dear Dr. ,Boydell

Thank you for submitting your manuscript to PLOS Global Public Health. After careful consideration, we feel that it has merit but does not fully meet PLOS Global Public Health’s publication criteria as it currently stands. Therefore, we invite you to submit a revised version of the manuscript that addresses the points raised during the review process.

We look forward to receiving your revised manuscript.

Kind regards,

Ateeb Ahmad Parray, BDS, MSS, MPH

Academic Editor

Journal Requirements:

Reviewers' comments:

Reviewer's Responses to Questions

**Comments to the Author**

1. If the authors have adequately addressed your comments raised in a previous round of review and you feel that this manuscript is now acceptable for publication, you may indicate that here to bypass the “Comments to the Author” section, enter your conflict of interest statement in the “Confidential to Editor” section, and submit your "Accept" recommendation.

Reviewer #3: All comments have been addressed

Reviewer #4: All comments have been addressed

Reviewer #5: (No Response)

2. Does this manuscript meet PLOS Global Public Health’s publication criteria? Is the manuscript technically sound, and do the data support the conclusions? The manuscript must describe methodologically and ethically rigorous research with conclusions that are appropriately drawn based on the data presented.

Reviewer #3: Yes

Reviewer #4: Yes

Reviewer #5: Yes

3. Has the statistical analysis been performed appropriately and rigorously?

Reviewer #3: N/A

Reviewer #4: Yes

Reviewer #5: Yes

4. Have the authors made all data underlying the findings in their manuscript fully available (please refer to the Data Availability Statement at the start of the manuscript PDF file)?

Reviewer #3: Yes

Reviewer #4: Yes

Reviewer #5: Yes

5. Is the manuscript presented in an intelligible fashion and written in standard English?

Reviewer #3: Yes

Reviewer #4: Yes

Reviewer #5: Yes

6. Review Comments to the Author

Reviewer #3: Authors have addressed comments

Reviewer #4: I congratulate the authors for a very important work done in a very intelligible and articulated manner. There are not additional comments but one which I am sure it was just a typo. On Page 13 there is a place where there was supposed to be a figure as t is mentioned 'insert figure 2'. I would recommend to double check for any such mistakes and proof read the manuscript before the final acceptance for publication.

Reviewer #5: "A systematic review of coercion and Long-Acting Reversible Contraceptive Methods (LARC)" is important for many reasons. This review is useful for academic and practical purposes.

LARC coercion is sensitive reproductive health issue. Uninformed reproductive decisions are coercion. Understanding LARC coercion's prevalence and impact is essential for reproductive autonomy and free contraceptive choice. This systematic review can illuminate the situation, identify risk factors, and suggest policies to eliminate coercion and protect reproductive rights.

Systematic reviews increase data quality. This systematic review analyses and synthesises coercion and LARC evidence. Methodological rigour eliminates bias and provides a thorough view of the issue. Thus, governments, clinicians, and researchers can trust the findings for informed decision-making, clinical practise, and further research.

Third, this extensive review enhances field scholarship. Reproductive health coercion demands constant research and analysis. Sharing this review's findings may encourage discussion, research collaborations, and a better understanding of coercion's sources and effects. This publication can encourage research on evidence-based therapies, policies, and guidelines to reduce coercion and enhance reproductive autonomy.

7. PLOS authors have the option to publish the peer review history of their article (what does this mean?). If published, this will include your full peer review and any attached files.

**Do you want your identity to be public for this peer review?** For information about this choice, including consent withdrawal, please see our Privacy Policy.

Reviewer #3: **Yes: **Apurva Kumar Pandya

Reviewer #4: **Yes: **Raafat Hassan

Reviewer #5: **Yes: **Muhammad Riaz Hossain

---

## [Editor Report · Decision Letter 2]

26 Jun 2023

Hidden in plain sight: A systematic review of coercion and Long-Acting Reversible Contraceptive Methods (LARC).

PGPH-D-23-00102R2

Dear Dr Boydell,

We are pleased to inform you that your manuscript 'Hidden in plain sight: A systematic review of coercion and Long-Acting Reversible Contraceptive Methods (LARC).' has been provisionally accepted for publication in PLOS Global Public Health.

Best regards,

Ateeb Ahmad Parray, BDS, MSS, MPH

Academic Editor

<quillbot-extension-portal></quillbot-extension-portal>